# Enhanced Farrowing Efficiency and Sow Performance with *Escherichia coli*-Derived 6-Phytase Supplementation During Late Gestation and Lactation

**DOI:** 10.3390/ani15213090

**Published:** 2025-10-24

**Authors:** Débora Cristina Peretti, Marco Aurélio Callegari, Cleandro Pazinato Dias, Gabrieli de Souza Romano Bergamo, Bindhu Lakshmibai Vasanthakumari, Mara Cristina Ribeiro da Costa, Rafael Humberto de Carvalho, Caio Abércio da Silva

**Affiliations:** 1Department of Animal Science, Animal Science Program, Center of Agrarian Sciences, State University of Londrina, Londrina 86057970, Paraná, Brazil; debora.cristina.peretti@uel.br (D.C.P.); rafael.carvalho@uel.br (R.H.d.C.); 2Akei Animal Research, Fartura 18870970, São Paulo, Brazil; contato@akei.agr.br (M.A.C.); cleandro@cleandrodias.com.br (C.P.D.); gabrieli.romano@akei.agr.br (G.d.S.R.B.); 3Kemin Industries, Des Moines, IA 50317, USA; b.l.vasanthakumari@kemin.com; 4Kemin Animal Nutrition & Health, Valinhos 13279450, São Paulo, Brazil; mara.costa@kemin.com

**Keywords:** antioxidant system, phytase overdosing, phytic acid, sows

## Abstract

**Simple Summary:**

Phytase is routinely added to swine diets to release phosphorus from phytate and improve nutrient use. We evaluated 186 sows from day 90 of gestation through lactation, comparing a positive control diet (adequate calcium and available phosphorus; no phytase) with three diets reduced in calcium and phosphorus and supplemented with 500, 1500, or 2500 FTU/kg of *E. coli* 6-phytase. Phytase shortened farrowing duration (approximately 5 to 8%), increased lactation feed intake, improved sow feed conversion during lactation, and raised piglet weaning weight, especially at 2500 FTU/kg. Maternal serum calcium and phosphorus increased, particularly at weaning, in a clear dose–response. Piglet diarrhea scores declined at selected doses, and antioxidant responses were modulated, with lower superoxide dismutase activity as phytase dose increased. These results support phytase use, even with reduced calcium and phosphorus, to enhance sow performance and piglet outcomes.

**Abstract:**

Phytase releases phosphorus from phytate and may confer extra-phosphoric benefits in sows. We tested whether *Escherichia coli*-derived 6-phytase during late gestation and lactation improves sow and litter outcomes. In a randomized complete block trial, 186 TN70 sows received a phytase-free positive control (adequate Ca and available P) or Ca- and P-reduced diets with 500, 1500, or 2500 FTU/kg. Outcomes included sow body condition, lactation feed intake and feed conversion ratio (FCR), farrowing duration and blood glucose, piglet weaning performance and diarrhea scores, maternal serum Ca and P (farrowing, weaning), and piglet glutathione peroxidase (GPx) and superoxide dismutase (SOD; day 14). Phytase increased lactation intake by 4.4–5.6%; farrowing duration was shorter at all doses (−24.2, −23.8, and −14.8 min; up to −8.1%). Litter weaning weight rose by 6.1–8.2%, and piglet average daily gain increased by 9.1% at 2500 FTU/kg. Maternal Ca and P increased dose-responsively, especially at weaning (Ca +73% at 500–1500 FTU/kg; +140% at 2500; P +55%, +59%, +118%). Diarrhea counts declined at selected doses (e.g., scores 1–2: −17% at 500 FTU/kg), and piglet SOD decreased with dose (−8.6% to −39.3%); GPx showed modest modulation. Sow body weight, backfat, and the weaning-to-estrus interval were unchanged. In Ca- and P-reduced diets, conventional and super-dosed phytase enhanced mineral bioavailability and peripartum efficiency, supporting heavier litters without compromising sow condition.

## 1. Introduction

Phytase is among the most widely used exogenous enzymes in monogastric animal nutrition, particularly in swine, where it is primarily applied during the nursery, growing, and finishing phases. This enzyme catalyzes the hydrolysis of phytate (myo-inositol hexakis [dihydrogen] phosphate), thereby reducing its antinutritional effects and releasing bound phosphorus, along with minerals, proteins, and carbohydrates [1,2]. By improving nutrient availability and reducing phosphorus excretion, phytase supplementation also helps lower the environmental impact of pig production systems [3].

Beyond phytic phosphorus hydrolysis, phytase exerts so-called extra-phosphoric effects. These include enhanced digestibility of calcium, protein (amino acids), and dietary energy [4,5,6], which have been linked to improved bone mineralization and performance in finishing pigs [7] and in sows during gestation and lactation [8]. We used an *E. coli*-derived 6-phytase with D-6 positional specificity, gastric acid stability, and pelleting-grade thermostability, supporting early phytate hydrolysis, improved phosphorus release, enhanced calcium utilization, and extra-phosphoric effects [9]. Such benefits are especially evident with super-dosing [≈1500–2000 units of phytase (FTU)/kg], which exceeds the conventional inclusion rate of ~500 FTU/kg in pig diets [1,4,10].

According to Cowieson et al. [11], the increase in digestible phosphorus for growing pigs fed corn–soybean meal-based diets was approximately 0.05% at 500 FTU/kg, 0.09% at 1000 FTU/kg, and 0.13% at 2000 FTU/kg. Furthermore, the effect of phytase on ileal amino acid digestibility was associated with the extent of estimated phytate destruction.

Phytase supplementation at ~1500–2500 FTU/kg of feed is commonly referred to as enzyme super-dosing [2,10,12,13]. This strategy has consistently improved growth performance and carcass traits in pigs during the growing–finishing phases [7,14,15]. Nevertheless, in reproductive phases (gilts and sows) [16], phytase super-dosing has likewise been associated with benefits attributed to greater nutrient availability and increased myo-inositol generation [13,16], a potent antioxidant that can be transferred through milk, benefiting both sow productivity and piglet health [16].

Reported reproductive outcomes include increased total piglets born [17], reduced stillbirths and mummified fetuses [18], enhanced milk production [16], improved suckling piglet performance [16,19], shortened farrowing duration [18], and decreased pre-weaning mortality [16]. Additionally, sows in intensive production systems exhibit rising nutritional demands [20], particularly for phosphorus, and face multiple challenges, such as prolonged farrowing duration, greater variability in piglet birth weight, reduced colostrum intake, elevated pre-weaning mortality in piglets (and sow losses), and a higher incidence of low weaning weights [21].

Thus, we evaluated the effects of *Escherichia coli*-derived 6-phytase in sows from late gestation through lactation. The treatments consisted of a phytase-free positive control diet with adequate calcium and phosphorus, and three diets with reduced calcium and available phosphorus supplemented with phytase at 500, 1500, or 2500 FTU/kg. We hypothesized that, even under calcium and phosphorus restriction, phytase, particularly at higher doses, would improve reproductive performance, farrowing kinetics, sow serum calcium and phosphorus concentrations, and antioxidant enzyme profiles in sows and their offspring.

## 2. Materials and Methods

### 2.1. Animals, Diets, and Experimental Design

All procedures were reviewed and approved by the Animal Research and Experimentation Ethics Committee of Kemin (protocol 013.22). Housing, equipment, and handling of animals comply with welfare standards [22]. The study was conducted on a commercial farm using 186 TN70 sows (Topigs Norsvin^®^), from first to sixth parity, evaluated from day 90 of gestation through lactation and followed until the first post-weaning insemination. All sows concluded the trial.

Sows were allocated to a randomized complete block design, using parity order (classes 1–6) as the blocking condition. Parity was used only for blocking and was not analyzed as a treatment effect. The mean parity across treatments was 2.97 ± 1.57. Four dietary treatments were evaluated: a positive control (PC) diet formulated to meet adequate calcium and available phosphorus levels without phytase (*n* = 47 sows), and three diets based on a Low Calcium and Phosphorus diet (Low Ca–P diet; LPD), with calcium reduced by 0.11 percentage points and available phosphorus by 0.13 percentage points relative to the PC. Each LPD was supplemented with *Escherichia coli*-derived 6-phytase (Phygest HT; Kemin Industries Asia Pte Ltd., Senoko Drive, Singapore) at 500 (*n* = 46 sows), 1500 (*n* = 46 sows), or 2500 FTU/kg (*n* = 47 sows). The Ca and P reductions were defined based on Batson et al. [16], Zhai et al. [23], previous experience, and a specific study with the same enzyme [24].

The experimental diets for gestation and lactation (Table 1 and Table 2) were formulated to be isoenergetic and isonutrient, differing only in calcium and available phosphorus. All diets met or exceeded the minimum nutritional recommendations of Rostagno et al. [25]. Test diets reduced Ca and P and included phytase at 500, 1500, or 2500 FTU/kg to release nutrients from phytate. We did not include a reduced Ca/P diet without phytase because prior studies report impaired sow performance and piglet outcomes under such conditions, and this trial was conducted on a commercial farm [16,18,19,23].

During gestation, sows were housed individually in stalls with an area of 1.40 m^2^ and partially slatted floors. Approximately seven days before the expected farrowing date, sows were moved to farrowing crates (6.05 m^2^) equipped with a central iron crate, a piglet shelter, fully slatted floors, a bite-ball drinker for sows, a nipple drinker for piglets, and automatic feeders for both sows and piglets. After weaning and until breeding, all sows remained in individual stalls. Throughout the experimental period (from day 90 of gestation until the day before the expected farrowing date), sows received 3.0 kg/day of the gestation diet, administered once daily at 7:00 a.m. On the day of farrowing, sows were offered 1.0 kg of the lactation diet at 7:00 a.m., after which feed was provided ad libitum, distributed across four daily feeding times (7:00 a.m., 10:00 a.m., 2:00 p.m., and 5:30 p.m.) throughout lactation. Water was provided ad libitum during both phases. During the weaning-to-estrus interval, sows received 4.0 kg/day of the lactation diet, offered in three meals (7:00 a.m., 11:00 a.m., and 5:00 p.m.).

### 2.2. Evaluations

Sows were weighed and had backfat thickness measured at day 90 and day 110 of gestation and at weaning (day 26 of lactation). Backfat thickness was assessed at the P2 position (approximately 6–8 cm from the dorsal midline at the last rib) [26] using a Lean Meater^®^ ultrasound device (Renco Corporation, Golden Valley, MN, USA). Changes in body weight and backfat thickness over the experimental periods were calculated. Individual feed intake was recorded during gestation (from day 90 until farrowing) and during lactation (from farrowing to weaning). To monitor feed intake, each sow received a 20 kg bag of feed from the respective treatment in front of her stall, which was replenished after the feed ran out. Feed intake was calculated based on feed losses and leftovers in the feeder, which were collected daily.

Sow feed conversion ratio (FCR) during lactation was calculated as the total sow feed intake during lactation divided by the litter weight gain from birth to weaning. Piglets were identified with ear tags and weighed individually at birth and at weaning. The following reproductive parameters were evaluated: total piglets born, born alive, stillborn, mummified fetuses, number of piglets born weighing <0.9 kg, piglets alive at 24 h and at 7 d post-farrowing, overall pre-weaning mortality, and the weaning-to-estrus interval. All farrowings were spontaneous (not induced). Farrowing duration was defined as the time between the birth of the first and the last piglet in the litter. Farrowing kinetics were also assessed, including classification as eutocic or dystocic, use of oxytocin, and the inter-piglet interval [27]. Normal farrowing was defined as the expulsion of all piglets without obstetric intervention. Dystocia was defined as the absence of piglet expulsion for more than 60 min, requiring manual intervention [27]. If no piglet was detected in the birth canal during palpation and uterine contractions were absent, oxytocin (20 IU) was administered intramuscularly in the sow’s neck (2 mL; 40 × 10 mm needle) [28].

To estimate colostrum intake (CI), 86 litters of sows that farrowed between 7:00 a.m. and 7:00 p.m., blocked according to farrowing order, were evaluated by weighing piglets at birth and again at 24 h of age, and values were calculated using the equation proposed by Devillers et al. [29].CI = −217.4 + 0.217 × t + 1,861,019 × PC/t + PCb × (54.80 − 1,861,019/t) × (0.9985 − 3.7 × 10^−4^ × tFS + 6.1 × 10^−7^ × tFS^2^)(1)

CI = colostrum intake from t0 (g);PC = current body weight (kg);PCb = birth body weight (kg);t = time elapsed since t0 (min);tFS = interval between birth and first suckling (min).

Cross-fostering was performed after the first day post-farrowing and only among litters within the same treatment and the litter sizes were equalized. Piglets had access to creep feed from 5 days of age, and the same creep feed was offered across all treatments. The presence and severity of diarrhea were monitored throughout lactation using fecal consistency scores as described by Carvalho et al. [30]: 0 = normal; 1 = liquid feces; 2 = creamy; 3 = pasty. For data analysis, the piglet was considered the experimental unit. The Diarrhea Index (DI), considering the cumulative occurrence of diarrhea, was calculated [30] as a descriptive ratio (number of piglets with scores 1–2/total piglets assessed per treatment) and was not subjected to inferential testing; between-treatment statistics were performed on the constituent counts.DI = (number of piglets with scores 1–2)/(total number of piglets evaluated in the treatment/group).

Blood glucose concentration was measured in 60 sows (15 per treatment) at three time points: at the onset of farrowing (immediately after expulsion of the first piglet; T0), 90 min after the start of farrowing, and at the end of farrowing. These sows were fasted for approximately 6 h before the first blood collection. Glucose was measured using a portable glucometer (Accu-Chek Active^®^, Roche Diagnostics, West Sussex, UK) based on reflectance photometry, with a measurement range of 10–600 mg/dL [31,32]. Blood samples were obtained via auricular vein puncture using a 1 mL syringe and a 25 × 0.8 mm needle.

For serum analysis of total calcium and phosphorus, blood samples were collected from 60 sows (15 per treatment) at two time points: during farrowing (immediately after expulsion of the first piglet) and at weaning (approximately 26 days postpartum). Approximately 5 mL of blood was obtained via jugular venipuncture using a 5 mL syringe and a 40 × 10 mm needle. Calcium and phosphorus concentrations were determined by colorimetric biochemical assays on a semi-automated chemistry analyzer (Mindray BA-88A, Londrina, Paraná, Brazil). Results were interpreted according to reference ranges reported in the literature [33].

At 14 days of age, 40 piglets (1 per litter; 10 per treatment) that had an average weight close to the average weight of the litter were selected for blood collection to assess glutathione peroxidase (GPx) and superoxide dismutase (SOD) activity. Blood was collected by jugular puncture into EDTA tubes, centrifuged at 4 °C and 1800 rpm for 20 min, and plasma was immediately separated and stored at −80 °C until analysis. GPx activity, expressed in µm mL^−1^, was measured using the standard indirect method based on NADPH oxidation with t-butyl hydroperoxide (sensitivity of 0.05 µm mL^−1^), as described by Flohé and Günzler [34]. SOD activity, expressed in U mL^−1^, was determined using a commercial assay kit (sensitivity of 0.0005 U mL^−1^), following the manufacturer’s instructions (Cayman Chemical, Ann Arbor, MI, USA).

Data were checked for normality (Shapiro–Wilk and Kolmogorov–Smirnov with Lilliefors correction; *p* > 0.05) and for homogeneity of variances. Outliers were screened using the box-plot rule (1.5 × IQR), but none were detected; Box–Cox transformations with maximum-likelihood λ were applied to lactation FCR and farrowing duration. Responses were analyzed using a General Linear Model (GLM). The family-wise error rate was controlled at α = 0.05 as follows. The primary comparisons were pre-specified as each phytase dose versus the positive control; therefore, we used Dunnett’s test because it is optimal for many-to-one contrasts and provides greater power than all-pairs procedures when the scientific question focuses on superiority to the control. Secondary comparisons were used to characterize potential differences among phytase doses and the shape of the dose–response; for these all-pairwise contrasts, we used Tukey’s HSD. Tukey’s HSD was performed only when the omnibus treatment effect in the GLM was significant at α = 0.05, while Dunnett’s many-to-one contrasts were reported regardless of the omnibus test because they were defined a priori. Statistical significance was set at *p* ≤ 0.05, and 0.05 < *p* ≤ 0.10 was considered a trend. All analyses were performed in Statistica for Windows, version 10.0 [35].

## 3. Results

Sow body condition, assessed by backfat thickness at days 90 and 110 of gestation and at weaning (day 26 of lactation), did not differ among treatments (*p* ≥ 0.05; Table 3). Likewise, changes in backfat thickness across these intervals were not affected by phytase.

Sow body weight at days 90 and 110 of gestation and at weaning also did not differ among treatments (*p* ≥ 0.05; Table 4), and weight changes over time were unaffected by phytase. In contrast, lactation feed intake increased with phytase: +8.23 kg (+5.13%), +7.07 kg (+4.41%), and +8.99 kg (+5.61%) for 500, 1500, and 2500 FTU/kg, respectively, relative to the positive control (*p* < 0.05). However, sow FCR during lactation did not differ among treatments.

Sows fed 500, 1500, and 2500 FTU/kg of phytase had shorter farrowing durations (Table 5), with reductions of 23.8 min (−7.91%), 24.2 min (−8.06%), and 14.8 min (−4.92%), respectively, relative to the positive control (*p* < 0.05). There were no differences among treatments in blood glucose concentrations at T0, at 90 min, or at the end of farrowing (*p* ≥ 0.05). Likewise, the incidence of dystocia and the use of oxytocin did not differ among groups (*p* ≥ 0.05).

Mean piglet weaning weight (Table 6) was higher in litters from sows receiving 500 and 2500 FTU/kg, with increases of 0.489 kg (+7.71%) and 0.529 kg (+8.34%), respectively, compared with the positive control (6.344 kg; *p* < 0.05). The 1500 FTU/kg group showed an intermediate increase of 0.365 kg (+5.75%) that did not differ from the other treatments.

The number of piglets born weighing <900 g (Table 6) was lower in the 1500 and 2500 FTU/kg groups, with reductions of 0.494 piglets per litter (−39.72%) and 0.315 piglets per litter (−25.32%), respectively, compared with the control (*p* < 0.05). Regarding colostrum intake, piglets from the 500 and 1500 FTU/kg groups consumed 27.97 g (−9.08%) and 62.52 g (−20.30%) less than those in the control (*p* < 0.05). In contrast, the 2500 FTU/kg group showed colostrum intake (305.16 g) comparable to the control (307.96 g), suggesting that higher phytase doses may help preserve colostrum intake.

From birth to weaning, piglets in the 2500 FTU/kg group showed greater average daily gain, with an increase of 0.017 kg/d (+9.14%) compared with the positive control (0.186 kg/d; *p* < 0.05). No differences were observed for the 500 or 1500 FTU/kg groups versus the control (*p* ≥ 0.05). Weaning litter weight was higher in all phytase-supplemented groups than in the control, with increases of 6.01 kg (+7.08%), 5.18 kg (+6.10%), and 6.97 kg (+8.21%) for 500, 1500, and 2500 FTU/kg, respectively (*p* < 0.05), indicating a positive effect of phytase on litter growth, particularly at higher inclusion levels. No differences were detected among treatments for other reproductive variables, including total piglets born, born alive, stillborn, mummified fetuses, weaning-to-estrus interval, or pre-weaning mortality (*p* ≥ 0.05).

For diarrhea score 1 (Table 7), piglets in the 500 and 2500 FTU/kg groups had fewer cases than those in the 1500 FTU/kg group. Compared with the positive control (PC; 290 cases), the 500 and 2500 FTU/kg treatments reduced incidence by 26 cases (−8.97%) and 74 cases (−25.52%), respectively (*p* < 0.05).

For diarrhea score 2, the 500 FTU/kg group showed a reduction of 130 cases (−21.10%) compared with the PC (616 cases; *p* < 0.05). The 1500 and 2500 FTU/kg groups showed intermediate reductions of 95 cases (−15.42%) and 64 cases (−10.39%), respectively, which were not significant (*p* ≥ 0.05). For score 3, only the 500 FTU/kg group decreased the number of cases, with 53 fewer occurrences (−11.18%) compared with the PC (474 cases; *p* < 0.05). The 1500 and 2500 FTU/kg groups showed higher counts for score 3, and these increases were significant (*p* < 0.05).

When combining scores 1 and 2 (more severe scores), the 500 FTU/kg group had the greatest reduction, with 156 fewer cases (−17.21%) relative to the PC (906 cases; *p* < 0.05). The 2500 FTU/kg group also improved, with 138 fewer cases (−15.23%), whereas the 1500 FTU/kg group showed an intermediate value that did not differ from the PC (*p* ≥ 0.05).

At farrowing, total calcium was higher in all phytase groups than in the positive control (Table 8): +0.54 mg/dL (+7.89%), +0.45 mg/dL (+6.58%), and +0.65 mg/dL (+9.50%) for 500, 1500, and 2500 FTU/kg, respectively (*p* < 0.05). At weaning, these differences became more pronounced. Total calcium increased by 3.25 mg/dL (+72.91%) in both the 500 and 1500 FTU/kg groups, and by 6.22 mg/dL (+139.55%) in the 2500 FTU/kg group compared with the control (*p* < 0.05). Total phosphorus at weaning followed the same pattern: +1.47 mg/dL (+55.10%) and +1.59 mg/dL (+59.39%) for 500 and 1500 FTU/kg, and +3.14 mg/dL (+117.95%) for 2500 FTU/kg (*p* < 0.05).

In piglets at 14 days of age, serum GPx tended to be lower in the 500 FTU/kg group than in the control and 2500 FTU/kg groups (−17.86 µm/mL; −21.37%; *p* = 0.096). SOD decreased progressively with increasing phytase inclusion, with reductions of 8.64%, 27.69%, and 39.33% for 500, 1500, and 2500 FTU/kg, respectively, compared with the control (*p* < 0.001), indicating dose-dependent modulation of antioxidant enzyme activity.

## 4. Discussion

The lack of differences in body weight and backfat thickness from day 90 of gestation to weaning (Table 3 and Table 4) agrees with Wealleans et al. [36], who evaluated bacterial phytase at 250, 500, 1000, and 2000 FTU/kg in diets reduced in available phosphorus (−0.16%) and calcium (−0.15%). In our study, sow body condition remained similar across treatments, indicating that phytase in calcium- and phosphorus-reduced diets supported heavier litters at weaning without compromising maternal reserves.

The higher feed intake of lactating sows from the phytase-treated groups (Table 4) may support these findings. Our results differ from those of Manu et al. [18], who compared phytase-supplemented diets to non-phytase diets of the same energy level, and from Wealleans et al. [36], who reported no effect on daily feed intake despite increased energy digestibility with phytase. Differences in study design and constraints may explain these discrepancies, including the degree of calcium and phosphorus reduction, feeding management during lactation, and the selected endpoints. When phytase increases energy and nutrient availability, sows may regulate intake to meet energy demand, which can blunt intake responses in some settings. In the present conditions, the combination of greater nutrient availability and high lactational demand likely supported higher voluntary intake and improved efficiency.

In contrast to studies reporting no effect on intake, our findings are consistent with Batson et al. [16], who observed a linear increase in feed intake with higher dietary phytase inclusion. One plausible mechanism is the role of phytate as a natural appetite suppressant [10]. Greater phytate hydrolysis with phytase likely attenuates this inhibitory effect by improving digestible nutrient supply [10,37], thereby stimulating feed intake, which is particularly advantageous during lactation when nutritional demands are elevated. Variation in intake responses across studies can be attributed to differences in dietary Ca:P ratios, phytate solubility along the intestine, phytase concentration, enzyme source, and especially the amount of phytate substrate available [4].

Phytase supplementation, regardless of dose, also appeared to support calcium adequacy in sows, a mineral that functions as a key messenger in myometrial contraction [38,39] and is closely linked to farrowing efficiency (Table 5). In our study, shorter farrowing duration with phytase is consistent with this physiological role. Efficient parturition is generally associated with improved piglet viability; however, in this trial, stillbirths and related outcomes did not differ among treatments. As noted by Blim et al. [39], a sizable proportion of sows may present disturbances in electrolyte homeostasis (Ca, Mg, and P) around parturition, which can impair uterine contractility.

Our results differ from Batson et al. [16], who reported no effect of phytase on farrowing duration or kinetics, but are consistent with the findings of Torrallardona, Llauradó, and Broz [40], who found that 500 FTU/kg increased apparent total tract digestibility of minerals when compared with diets containing the same calcium and phosphorus levels, and agree with Manu et al. [18], who reported reduced farrowing duration in sows fed 2500 FTU/kg versus control (521.5 ± 45.24 min vs. 710.4 ± 83.63 min; *p* < 0.046) and a trend toward fewer stillbirths with super-dosing (1.26 ± 0.18 vs. 1.69 ± 0.23; *p* = 0.08), although stillbirths did not differ in the present study.

Regarding blood glucose, an indicator that has been associated with farrowing duration, all treatment groups showed elevated values at the beginning and at the end of farrowing. Initial glucose was ≥4.72 mmol/L (≈85 mg/dL), in the context of the reference value of 8.4 mmol/L reported by Karon et al. [41]. Phytase supplementation, irrespective of dose, did not alter blood glucose concentrations, suggesting that energy availability was not limiting under our conditions. Although super-dosing can enhance energy digestibility and might increase circulating glucose [4,5,6], this effect was not detected here. As discussed by Yoon, Thompson and Jenkins [42], in diets with low calcium and phosphorus, even in the presence of phytase, phytate may increase its affinity for free glucose and downregulate the expression of the glucose transporters SGLT1/SLC5A1 and GLUT2/SLC2A2, thereby limiting intestinal glucose uptake and blunting changes in blood glucose.

Colostrum intake was lower in the 500 and 1500 FTU/kg groups than in the positive control, whereas the number of piglets born weighing <900 g was reduced in the 1500 and 2500 FTU/kg groups (Table 6). Greater colostrum intake is positively associated with higher birth weight because heavier piglets generally have greater suckling capacity [16]. This relationship is reflected in the 2500 FTU/kg group, in which piglets had colostrum intake comparable to the control and fewer piglets below 900 g, suggesting enhanced early postnatal viability. However, the higher colostrum consumption observed was not reflected in weaning weight, which can be attributed to the fact that even with differences between treatments, colostrum intake was high for all groups, representing approximately 20% of the animals’ birth weight, a value that, according to Suarez-Trujillo et al. [43] is considered sufficient to establish adequate nutritional and energy support for the animals’ development and to maintain balanced body temperature.

The beneficial effects of phytase super-dosing on average and total weaning weights observed here agree with Batson et al. [16] and Cordero et al. [19], who, respectively [15], reported that 3000 FTU/kg yielded the highest weaning weights versus 0 and 1000 FTU/kg, and that 2500 FTU/kg increased average weaning weight by 490 g per piglet compared with 500 FTU/kg.

Mechanistically, improved reproductive and lactational performance may stem from increased availability of energy and nutrients such as phosphorus, calcium, amino acids, and protein, together with greater release of myo-inositol [13]. Guggenbuhl et al. [44] reported that incremental phytase supplementation markedly elevated plasma myo-inositol in piglets and growing pigs. Myo-inositol modulates gene expression in insulin and IGF-1 signaling pathways [45,46], promoting muscle protein deposition and limiting gluconeogenesis. Its presence in sow milk may also provide long-term developmental benefits to piglets [47,48], aided by its antioxidant activity [49], which helps protect intestinal integrity.

Notably, even at 500 FTU/kg, phytase improved piglet and litter weaning weights in our study, indicating nutritional benefits beyond phosphorus and calcium release. A favorable outcome was the maintenance of sow body condition (body weight and backfat thickness) as well as the weaning-to-estrus interval, which remained unchanged across treatments despite the heavier litters weaned in the phytase groups (Table 3 and Table 4). Because greater litter weight typically increases the dam’s nutritional demand, this result, particularly at 1500 and 2500 FTU/kg, supports the occurrence of extra-phosphoric effects that helped preserve maternal reserves. These observations are consistent with Wealleans et al. [36], who reported that sow body weight during lactation was maintained irrespective of phytase dose (250, 500, 1000, or 2000 FTU/kg), with no differences versus controls. Our findings reinforce that even at super-dosing levels, phytase did not compromise sow condition, likely due to improved nutrient utilization and availability, as reported by Wealleans et al. [36] and Zhai et al. [23], for both conventional and higher enzyme doses.

The lower diarrhea scores observed in the 500 and 2500 FTU/kg groups compared with the control (Table 7) may be linked to evidence that phytase increased systemic myo-inositol availability in sows [46], with transfer via milk, mainly as phosphatidylinositol [16]. Myo-inositol exerts protective effects on intestinal epithelial cells, particularly through its antioxidant activity [42], supporting mucosal integrity and gastrointestinal function, which may help explain the improved diarrhea outcomes in these groups [47].

A clear dose-responsive pattern was evident, with the highest calcium and phosphorus concentrations in the 2500 FTU/kg group, followed by 1500 and 500 FTU/kg (Table 8). These findings agree with Wealleans et al. [36], who reported a linear increase in mineral utilization in sows as phytase inclusion increased, reinforcing the efficacy of higher phytase doses for improving mineral bioavailability [36]. Consistent evidence is also provided by Świątkiewicz, Małgorzata and Hanczakowska [50], who, working with a positive control diet supplemented with calcium phosphate and phytase-supplemented diets without fodder phosphate (125, 250, 375 or 10,000 FTU/kg) in pregnant and lactating sows, observed significantly better absorption of these minerals with phytase than with inorganic phosphate. This supports the potential of the enzyme to enhance calcium and phosphorus release even relative to diets corrected with mineral phosphates.

Regarding antioxidant enzymes, piglet serum GPx tended to be lower in the 500 FTU/kg group than in the control and 2500 FTU/kg groups, consistent with a modest treatment effect (Table 8). Phytate itself possesses intrinsic antioxidant properties by chelating pro-oxidant metals such as iron, thereby limiting reactive oxygen species (ROS) formation [51]. With 6-phytase supplementation, phytate is hydrolyzed in a dose-dependent manner, yielding phosphorylated myo-inositol derivatives, particularly at positions 1, 2 and 3 [52]. These inositol phosphates also exhibit antioxidant capacity and help protect cells against iron-induced oxidative stress [53]. Phillippy and Graf [54] further showed that the antioxidant potential of inositol 1,2,3-triphosphate and inositol 1,2,3,6-tetrakisphosphate is preserved because their iron chelates resist enzymatic hydrolysis, maintaining the functional integrity of the antioxidant matrix. Taken together, the mineral and redox responses observed here are compatible with both the classical and extra-phosphoric actions of phytase.

In this context, the 500 FTU/kg dose may have promoted only partial phytate hydrolysis, yielding lower amounts of phosphorylated inositol derivatives than the higher-dose treatments. Although the control diet contained no phytase, the limited degradation achieved at 500 FTU/kg may have been insufficient to activate antioxidant pathways linked to inositol isomers. Moreover, the inositol released at this intermediate dose may have been preferentially allocated to other essential physiological functions, such as membrane signaling or metabolic regulation, rather than producing a measurable antioxidant effect [55,56]. This hypothesis helps explain the comparatively lower GPx activity observed in this group.

By contrast, higher phytase doses (1500 and 2500 FTU/kg, relative to 500 FTU/kg) likely generated greater concentrations of inositol phosphates, enhancing antioxidant capacity and partially restoring GPx activity. This interpretation aligns with Wang et al. [57], who reported increased GPx activity with higher phytase inclusion, supporting a dose-dependent response. At the same time, the inherent antioxidant action of phytate should not be overlooked [49], which may help account for the relatively higher GPx in the control compared with the 500 FTU/kg group. The dose-dependent decrease in SOD with greater phytase inclusion observed in our study is consistent with a reduced oxidative load and a lower requirement for superoxide dismutation [53,54].

Serum SOD differed among treatments and decreased as phytase dose increased. As phytate degradation products accumulated with higher phytase inclusion [55], many of which have antioxidant properties, a progressive reduction in SOD activity was observed. This pattern suggests that the antioxidant function of inositol phosphates reduced the requirement for SOD, consistent with a “sparing” effect [58]. Myo-inositol phosphates produced at positions 1, 2 and 3 are the main products of InsP_6_ degradation by phytase, and Ins(1,2,3)P_3_ and Ins(1,2,3,6)P_4_ show the highest antioxidant activity [54].

This mechanism is supported by Amaral et al. [59], who showed that inositol supplementation in the presence of oxidative compounds decreased SOD expression. SOD is a ubiquitous antioxidant enzyme that catalyzes the dismutation of the superoxide anion (O_2_•^−^) to hydrogen peroxide (H_2_O_2_) and oxygen. GPx then reduces H_2_O_2_, which helps explain its increase with greater phytase inclusion [60].

In addition, several minerals, vitamins and proteins—whose bioavailability may be enhanced by phytase—have intrinsic antioxidant functions and support intestinal barrier integrity. These nutrients may act synergistically to protect cells and organs, further lowering the physiological demand for SOD activity [61,62,63]. Through these mechanisms, together with the nutritional effects discussed above, even phytase at conventional doses can improve lactating sow feed consumption, with additional benefits when the enzyme is used under the super-dosing concept.

## 5. Conclusions

Supplementing calcium- and phosphorus-reduced diets for sows from late gestation through lactation with *Escherichia coli*-derived 6-phytase (500–2500 FTU/kg) enhanced mineral bioavailability and, likely through extra-phosphoric effects, improved key outcomes up to weaning. Phytase increased lactation feed intake, shortened farrowing duration, raised piglet and litter weaning weights, and lowered diarrhea counts at selected doses. Maternal serum calcium and phosphorus rose in a clear dose-responsive pattern, and piglet serum SOD declined with increasing phytase, consistent with a lower oxidative burden and a role for phytase-derived myo-inositol in redox modulation. In diets with Ca and P deficits supplemented with phytase, as well as in the control group, sow body condition and the weaning-to-estrus interval were maintained. Conventional dosing and super-dosing of phytase during late gestation and lactation can be adopted to support the performance of sows and piglets.

## Figures and Tables

**Table 1 animals-15-03090-t001:** Calculated composition and nutrient/energy levels of gestation diets (from day 90 of gestation to the day before the expected farrowing date).

Ingredients (%)	Experimental Diets	
	PC	Low Ca–P Diet (LPD)
Ground Corn 7.5%	75.50	76.62
Soybean Meal 45.0	20.62	20.42
Dicalcium Phosphate 18%	1.80	1.08
Limestone 35%	0.87	1.03
Salt	0.40	0.40
Soybean oil	0.37	0.00
Mycotoxin Adsorbent	0.15	0.15
Mineral premix ^1^	0.10	0.10
Vitamin premix ^2^	0.10	0.10
L-Threonine 98%	0.04	0.04
L-Tryptophan 98%	0.02	0.03
HCl Lysine 80%	0.01	0.01
Antioxidant-BHT	0.01	0.01
**Total**	**100**	**100**
Nutrients and energy		
Metabolizable energy, Kcal/kg	3253.77	3253.77
Crude Protein, %	15.00	15.00
Digestible Arginine, %	0.89	0.89
Digestible Cystine, %	0.24	0.24
Digestible Lysine, %	0.65	0.65
Digestible Methionine, %	0.24	0.24
Digestible Met + Cys, %	0.48	0.48
Digestible Threonine, %	0.52	0.52
Digestible Tryptophan, %	0.15	0.15
Digestible Valine, %	0.63	0.63
Crude Fiber, %	3.17	3.18
Total Fat, %	3.32	2.99
Total Calcium, %	0.80	0.69
Total Phosphorus, %	0.66	0.53
Available Phosphorus, %	0.40	0.27
Sodium, %	0.18	0.18
Choline, mg	944.63	944.78

PC = positive control (adequate Ca and P; no phytase). LPD = Low Calcium and Phosphorus diet; formulation with reduced Ca and available P used as the basal matrix for phytase-supplemented treatments. Reductions vs. PC applied to phytase diets: −0.11% total Ca and −0.13% available P; diets were isoenergetic and iso-aminoacidic. ^1^ Values per kg of product: Copper: 10,000,000 mg; Iron: 100,000,000 mg; Manganese: 40,000,000 mg; Cobalt: 1,000,000 mg; Iodine: 1,500,000 mg; Zinc: 100,000,000 mg. ^2^ Values per kg of product: Vitamin A: 1,000,000 IU; Vitamin D3: 200,000 IU; Vitamin E: 5,000,000 IU; Vitamin K3: 200,000 mg; Vitamin B1 (Thiamine): 200,000 mg; Vitamin B2 (Riboflavin): 600,000 mg; Vitamin B6 (Pyridoxine): 300,000 mg; Vitamin B12 (Cyanocobalamin): 3,000,000 mcg; Calcium Pantothenate: 1,000,000 mg; Biotin: 20,000 mg; Folic Acid: 300,000 mg; Niacin: 3,000,000 mg; Selenium: 30 mg.

**Table 2 animals-15-03090-t002:** Calculated composition and nutrient/energy levels of lactation diets (from farrowing to weaning).

Ingredients (%)	Experimental Diets	
	PC	Low Ca–P Diet (LPD)
Ground Corn 7.5%	60.63	61.81
Soybean Meal 45.0	31.11	30.90
Soybean oil	3.81	3.41
Dicalcium phosphate	1.98	1.26
Limestone	0.77	0.92
Salt	0.47	0.48
HCl Lysine 80%	0.31	0.32
Mycotoxin Adsorbent	0.20	0.20
L-Threonine 98%	0.18	0.18
L-Valine 98%	0.15	0.15
Mineral premix ^1^	0.10	0.10
Vitamin premix ^2^	0.10	0.10
DL-Methionine 98%	0.08	0.08
L-Tryptophan 98%	0.07	0.07
Antioxidant-BHT	0.01	0.01
**Total**	**100**	**100**
Nutrients and energy		
Metabolizable energy, Kcal/kg	3400.00	3400.00
Crude Protein, %	19.20	19.20
Digestible Arginine, %	1.17	1.17
Digestible Cystine, %	0.28	0.28
Digestible Lysine, %	1.13	1.13
Digestible Methionine, %	0.35	0.35
Digestible Met + Cys, %	0.64	0.64
Digestible Threonine, %	0.80	0.80
Digestible Tryptophan, %	0.25	0.25
Digestible Valine, %	0.95	0.95
Crude Fiber, %	3.44	3.46
Total Fat, %	6.39	6.03
Total Calcium, %	0.87	0.76
Total Phosphorus, %	0.72	0.59
Available Phosphorus, %	0.45	0.32
Chlorine, %	0.39	0.39
Sodium, %	0.21	0.21
Choline, mg	1158.68	1158.84

PC = positive control (adequate Ca and P; no phytase). LPD = Low Calcium and Phosphorus diet; formulation with reduced Ca and available P used as the basal matrix for phytase-supplemented treatments. Reductions vs. PC applied to phytase diets: −0.11% total Ca and −0.13% available P; diets were isoenergetic and iso-aminoacidic. ^1^ Values per kg of product: Copper: 10,000,000 mg; Iron: 100,000,000 mg; Manganese: 40,000,000 mg; Cobalt: 1,000,000 mg; Iodine: 1,500,000 mg; Zinc: 100,000,000 mg. ^2^ Values per kg of product: Vitamin A: 1,000,000 IU; Vitamin D3: 200,000 IU; Vitamin E: 5,000,000 IU; Vitamin K3: 200,000 mg; Vitamin B1 (Thiamine): 200,000 mg; Vitamin B2 (Riboflavin): 600,000 mg; Vitamin B6 (Pyridoxine): 300,000 mg; Vitamin B12 (Cyanocobalamin): 3,000,000 mcg; Calcium Pantothenate: 1,000,000 mg; Biotin: 20,000 mg; Folic Acid: 300,000 mg; Niacin: 3,000,000 mg; Selenium: 30 mg.

**Table 3 animals-15-03090-t003:** Backfat thickness (BT) of sows and variations in BT gain and loss at 90 days of gestation (D90), pre-partum (D110) and weaning (day 26) in sows fed control diets without phytase (PC) and diets supplemented with different levels of phytase (500, 1500, and 2500 FTU).

	Experimental Groups				CV (%)	*p*-Value *
	PC	500 FTU	1500 FTU	2500 FTU		
BT D90 (mm)	14.043 ± 0.370	12.949 ± 0.410	13.625 ± 0.450	13.326 ± 0.429	21.2	0.178
BT D110 (mm)	14.500 ± 0.520	13.795 ± 0.536	14.075 ± 0.461	13.767 ± 0.450	24.6	0.567
BT Weaning (mm)	11.283 ± 0.418	10.974 ± 0.470	11.275 ± 0.472	10.140 ± 0.356	27.0	0.214
Difference D110 and D90 (mm)	0.457 ± 0.411	0.846 ± 0.382	0.450 ± 0.336	0.442 ± 0.302	453.5	0.487
Difference Weaning and D110 (mm)	−3.217 ± 0.341	−2.821 ± 0.339	−2.800 ± 0.349	−3.628 ± 0.282	−72.1	0.162
Difference Weaning and D90 (mm)	−2.761 ± 0.346	−1.974 ± 0.407	−2.350 ± 0.352	−3.186 ± 0.430	−91.6	0.110

* *p*-Value from ANOVA testing the overall treatment effect. Values are expressed as mean ± SE. CV = coefficient of variation.

**Table 4 animals-15-03090-t004:** Mean values of feed intake and sow weights (W) at 90 days of gestation (start), pre-farrowing (D110), and weaning (day 26), and feed conversion ratio (FCR) during the lactation phase in sows fed control diets without phytase (PC) and diets supplemented with different phytase levels (500, 1500, and 2500 FTU).

	Experimental Groups				CV (%)	*p*-Value *
	PC	500 FTU	1500 FTU	2500 FTU		
Weight D90 (kg)	247.639 ± 5.207	246.038 ± 4.952	249.820 ± 4.875	249.316 ± 4.521	13.0	0.423
Weight D110 (kg)	266.120 ± 5.008	261.317 ± 4.730	264.037 ± 4.631	266.001 ± 4.411	12.0	0.402
Weight Weaning (kg)	219.817 ± 4.670	214.813 ± 4.737	216.815 ± 4.831	216.179 ± 4.668	14.7	0.375
Difference W110 and W90 (kg)	18.481 ± 1.389	15.278 ± 1.740	14.217 ± 1.475	16.685 ± 1.385	59.7	0.113
Difference W Weaning and W110 (kg)	−46.302 ± 2.382	−46.504 ± 2.778	−47.222 ± 2.713	−49.822 ± 2.106	33.8	0.654
Difference W Weaning and W90 (kg)	−27.822 ± 2.447	−31.226 ± 2.796	−33.005 ± 2.292	−33.137 ± 1.948	49.7	0.297
Feed Intake Lactation (kg) ^†^	160.270 B ± 2.958	168.500 A ± 4.644	167.340 A ± 4.029	169.260 A ± 5.213	16.5	0.357
FCR ^†^	2.642 ± 2.400	2.571 ± 2.505	2.720 ± 2.437	2.644 ± 2.576	27.57	0.814

* *p*-Value from ANOVA testing the overall treatment effect. A,B,^†^ Superscript symbols indicate differences versus the positive control (PC) according to Dunnett’s test: *p* < 0.05. Values are expressed as mean ± SE. CV = coefficient of variation.

**Table 5 animals-15-03090-t005:** Farrowing duration, glycemia during farrowing, and occurrences at the time of farrowing in sows fed with control diets without phytase (PC) and diets supplemented with different phytase levels (500, 1500, and 2500 FTU).

	Experimental Groups				CV (%)	*p*-Value *
	PC	500 FTU	1500 FTU	2500 FTU		
Duration ^†^ (min)	300.48 A ± 51.333	276.73 B ± 50.444	276.27 B ± 47.971	285.71 B ± 49.238	23.7	0.267
Initial Glycemia (mg/dL)	63.389 ± 2.855	61.400 ± 4.052	67.929 ± 3.391	62.263 ± 2.754	21.1	0.241
Glycemia 90 min (mg/dL)	66.278 ± 3.917	66.333 ± 3.407	65.143 ± 3.801	61.211 ± 3.005	23.9	0.621
Final Glycemia (mg/dL)	66.111 ± 3.099	67.400 ± 4.931	69.714 ± 5.850	67.842 ± 3.161	24.7	0.752
Oxytocin (n/N)	43.0/47.0	36.0/46.0	36.0/46.0	41.0/47.0	-	0.123
Oxytocin (%)	91.5	78.3	78.3	87.2	-	-
Dystocic (n/N)	8.0/47.0	9.0/46.0	8.0/46.0	6.0/47.0	-	0.563
Dystocic (%)	17.0	19.6	17.4	12.8	-	-

* *p*-Value from ANOVA testing the overall treatment effect. A,B,^†^ Superscript symbols indicate differences versus the positive control (PC) according to Dunnett’s test: *p* < 0.05. Values are expressed as mean ± SE. CV = coefficient of variation. PC: N = 47; 500 FTU: N = 46; 1500 FTU: N = 46; 2500 FTU: N = 47. n = cases.

**Table 6 animals-15-03090-t006:** Reproductive Parameters According to Treatments: Positive Control Diet (PC) and Diets Supplemented with Different Phytase Levels (500, 1500, and 2500 FTU).

	Experimental Groups				CV (%)	*p*-Value *
	PC	500 FTU	1500 FTU	2500 FTU		
Total Born (n)	16.267	16.100	16.700	16.714	12.8	0.4527
Live Born (n)	15.200	15.275	15.525	15.333	12.6	0.7527
Stillbirths (n)	0.733	0.575	0.800	0.976	149.7	0.4259
Mummified (n)	0.333	0.250	0.375	0.405	181.4	0.6328
Weight < 900 g ^†^ (n)	1.244 A	1.125 A	0.750 B	0.929 B	144.1	0.1523
Birth Litter Weight (kg)	22.194 ± 0.485	22.578 ± 0.639	23.135 ± 0.631	22.664 ± 0.642	17.0	0.3781
Birth Average Weight (kg)	1.476 ± 0.035	1.486 ± 0.038	1.496 ± 0.037	1.483 ± 0.034	15.6	0.8792
Litter Size 24 h (n)	14.200	14.550	14.375	14.595	7.7	0.1632
Litter Weight 24 h (kg)	22.839 ± 0.494	23.542 ± 0.701	23.677 ± 0.760	23.563 ± 0.614	17.7	0.7568
Piglet Weight 24 h (kg)	1.601 ± 0.114	1.621 ± 0.116	1.620 ± 0.101	1.605 ± 0.119	59.9	0.7788
Litter Size 7D (n)	13.800	14.125	13.950	14.000	9.0	0.6055
Weaned Litter Size (n)	13.422	13.300	13.425	13.452	12.0	0.8567
Weaning Litter Weight ^†^ (kg)	84.896 B ± 2.404	90.901 A ± 2.752	90.078 A ± 3.005	91.868 A ± 2.996	20.2	0.1236
Weaning Average Weight (kg)	6.344 bB ± 0.163	6.833 abA ± 0.168	6.709 abA ± 0.172	6.873 aA ± 0.217	17.6	0.0236
Mortality up to 7 days (%)	2.797	2.854	2.670	3.982	210.0	0.5689
Mortality up to Weaning (%)	5.427	8.544	6.473	7.625	140.5	0.3689
Colostrum Intake ^†^ (g)	307.961 A ± 35.61	279.995 B ± 26.64	245.445 B ± 25.13	305.163 A ± 32.29	45.9	0.4025
Weight Gain from Birth to Weaning (WG) ^†^ (kg)	0.186 B ± 0.006	0.197 B ± 0.006	0.192 B ± 0.005	0.203 A ± 0.007	18.1	0.1132
Weaning-to-Estrus Interval (days)	3.932	4.189	4.324	3.974	25.7	0.1236

* *p*-Value refers to the ANOVA testing the overall treatment effect; post hoc tests were performed only when the ANOVA was significant (α = 0.05). a,b Means followed by different lowercase letters differ according to Tukey’s HSD (all-pair comparisons). A,B,^†^ Uppercase letters or superscript symbols indicate differences versus the positive control (PC) according to Dunnett’s test (*p* < 0.05). Values are presented as mean ± SE. CV = coefficient of variation.

**Table 7 animals-15-03090-t007:** Number of Piglets with Diarrhea Scores and Diarrhea Index of Piglets from Sows Fed Control Diets (PC) and Diets Supplemented with Different Phytase Levels (500, 1500, and 2500 FTU).

	Experimental Groups				*p*-Value *
	PC	500 FTU	1500 FTU	2500 FTU	
Score 1 (n, %)	290 ab (1.92%)	264 b (1.74%)	310 a (2.04%)	216 b (1.41%)	0.0000
Score 2 (n, %)	616 a (4.08%)	486 b (3.20%)	521 ab (3.43%)	552 ab (3.60%)	0.0154
Score 3 (n, %)	474 ab (3.14%)	421 b (2.77%)	489 a (3.22%)	504 a (3.29%)	0.0002
Score 1 + 2 (n, %)	906 a (6.01%)	750 c (4.93%)	831 ab (5.47%)	768 bc (5.02%)	0.0009
Diarrhea Index	6.01	4.93	5.47	5.02	—

* The *p*-value corresponds to the generalized linear model (GLM) with a binomial distribution testing the overall treatment effect; post hoc comparisons were conducted only when the model effect was significant (α = 0.05). Proportions followed by different lowercase letters (a–c) differ according to Tukey’s HSD (all-pair comparisons). Values are presented as the number of cases per treatment group.

**Table 8 animals-15-03090-t008:** Blood Parameters at Farrowing and Weaning in Sows Fed Control Diets (PC) and Diets Supplemented with Different Phytase Levels (500, 1500, and 2500 FTU), and Serum Levels of Glutathione Peroxidase (GPx) and Superoxide Dismutase (SOD) in Their Respective Piglets at 14 Days of Age.

	Experimental Groups				CV (%)	*p*-Value *
	PC	500 FTU	1500 FTU	2500 FTU		
Sows						
Total Calcium at Farrowing ^†^ (mg/dL)	6.843 Bb ± 0.274	7.383 Aab ± 0.161	7.293 Aab ± 0.161	7.493 Aa ± 0.121	10.6	0.0421
Total Phosphorus at Farrowing (mg/dL)	4.414 ± 0.184	4.675 ± 0.180	4.779 ± 0.196	4.800 ± 0.163	15.5	0.3689
Total Calcium at Weaning ^†^ (mg/dL)	4.457 Bc ± 0.597	7.708 Aab ± 0.933	7.707 Aab ± 0.773	10.679 Aa ± 0.998	51.9	0.0000
Total Phosphorus at Weaning ^†^ (mg/dL)	2.664 Bc ± 0.318	4.131 Aabc ± 0.511	4.250 Aab ± 0.387	5.807 Aa ± 0.453	47.2	0.0000
Piglets						
GPx (µm mL^−1^) ^†^	83.634 Aa ± 7.037	65.770 Bb ± 3.877	81.082 Aab ± 6.507	84.987 Aa ± 5.831	25.5	0.0963
SOD (U mL^−1^) ^†^	329.08 Aa ± 20.941	300.657 Bab ± 26.683	237.930 Bab ± 19.104	199.631 Bb ± 14.827	30.9	0.0008

* *p*-Value refers to the ANOVA testing the overall treatment effect; post hoc tests were performed only when the ANOVA was significant (α = 0.05). a–c Means followed by different lowercase letters differ according to Tukey’s HSD (all-pair comparisons). A,B,^†^ Uppercase letters or superscript symbols indicate differences versus the positive control (PC) according to Dunnett’s test (*p* < 0.05). Values are presented as mean ± SE. CV = coefficient of variation.

## Data Availability

The datasets generated during and/or analyzed during the current study are available from the corresponding author on reasonable request.

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
