# Peer review of "Enhanced Farrowing Efficiency and Sow Performance with Escherichia coli-Derived 6-Phytase Supplementation During Late Gestation and Lactation"

_animals, 2025, doi:10.3390/ani15213090_

Round 1

Reviewer 1 Report

Comments and Suggestions for Authors

Dear Authors, 
Thank you for the opportunity to review this manuscript. The study addresses an interesting topic and has potential value, but some points need clarification and improvement before it can be considered for publication. I would like you to know that I have added various notes to the attached PDF document.

Line 92: What is the purpose of block factor grouping? Where are the results of these block groups presented? Where are the effects on trial results shown and discussed?

Line 96: The findings are not clear on whether the effect of these three diets compared to the positive control stems from changes in calcium and phosphorus ratios or from the addition of phytase. The diet treatment referred to as the negative control (NC) (althouth it is not stated exactly in the text)  without phytase should also have been examined.

Line 103: In addition to the previous review on the phytate-free negative control diet, there are many studies in the literature regarding the reduction of calcium and phosphorus in the diet. While this restriction is unlikely to cause the problems in the animals, it also makes it impossible to compare the results.

Line 105, Table 1: (NC): Negative Control? Please give this words.

Total CA and Total P: Is there a reference for the reduction amount? How will we know that a sufficient reduction has been made to enable comparison?

Line 142: How it was recorded can be briefly explained

Line 176: The reference cited (and the references cited by that reference) do not clearly base this index formula on scientific reality. This formula represents a ratio rather than an index value. As the findings were not statistically analyzed, the discussion does not reflect scientific reality.

Line 221:Table 3 title shoul be simplified.

CV (%): Why was the coefficient of variation value used?

BT D90 (mm) : What does this abbreviation mean? In same line the standard error of the mean must be used for each compare, each table.

Line 230: This statement is only correct for the 500 FTU group. Table 4 shows that the differences in other groups are insignificant.

Table 4: How do you explain that feed intake was higher in all treatment groups, yet FCR statistically improved only at 500 FTU and the other treatments were similar in statistically? Considering this fact, changes will need to be made in the Discussion section.

Table 6: Considering the similarities in average birth weights, it should be explain why the number of piglets under 900 g is higher in the PC and 500 FTU groups. To understand this more clearly, it is essential to provide standard errors. (W, BLW and BAW)

Table 6: It should be explained why piglets in the PC group, despite having similar birth weights and higher colostrum intake, had significantly lower weaning weights at 500 and 1500 FTU. (WAW)

Line 293, Table 7 : Where is this data present in the table?

Diarrhea Index: Since there is no statistical analysis for this index (or, more accurately, “percentage”) value, it is not scientifically possible to understand it.

Line 309: (For p value) In general, it is confusing that the P-value given in the tables and the Tukey and Dunnet tests given below the table are provided separately. It is unclear and confusing which test we should consider and when.

Line 323: According to Table 6, the Weaning Average Weight of piglets had only increased by 2500 FTU. This statement is incorrect.

Line 326: unsuitable first sentence for the beginning of a new paragraph

Line 332: The parity structure was not sufficiently explained and the findings were not presented.

Line 417: The relevant datas are insufficient to suggest this claim. This information is an unrealistic assumption.

Line 423: Although there are no clear results indicating that the treatment improved diarrhea, using this statement is far from reality. Statistically, only 500 FTU appears to have improved in Score 2, while the 1500 and 2500 FTU groups are similar to PC. On the other hand, similar results to 1500 FTU were obtained in Score 1+2.

Line 432: All these discussions have remained unresolved due to the missing phytase-free negative control group. It can be taken as an observation and comment, but it is not sufficient for scientific reality.

Line 487: This assumption is clearly incorrect based on litter size datas.

Line 498: What does this statement mean? Does it mean “not maintained” in the PC group? If so, it is a statement far from scientific reality, as the differences in BT and the weaning-to-estrus interval values are similar between all treatment groups and the PC group.

Line 499: In general, it cannot be stated that the findings support this use. However, the authors may recommend its use.

Author Response

REVIEWER 1

Dear Authors,

REV01: Thank you for the opportunity to review this manuscript. The study addresses an interesting topic and has potential value, but some points need clarification and improvement before it can be considered for publication. I would like you to know that I have added various notes to the attached PDF document.

Dear Reviewer,

Thank you for your thorough review and for the annotated notes. We revised the manuscript to improve clarity in Methods (blocking, intake recording, DI definition), tables/footnotes (SE vs CV; test labels; titles), consistency between text and tables (e.g., FCR), and the scope and strength of claims in the Discussion and Conclusions. These revisions are reflected in the marked file.

REV01: Line 92: What is the purpose of block factor grouping? Where are the results of these block groups presented? Where are the effects on trial results shown and discussed?

AUT# Lines 93 - 94: Sows were blocked by parity to control predictable variability in reproductive performance across parities, thereby increasing precision of treatment comparisons. Blocking was not a factor of scientific interest and therefore was not tested nor reported as an effect; it served solely to reduce unexplained variance so that treatment effects could be estimated more precisely. We clarified this in Methods.

REV01: Line 96: The findings are not clear on whether the effect of these three diets compared to the positive control stems from changes in calcium and phosphorus ratios or from the addition of phytase. The diet treatment referred to as the negative control (NC) (althouth it is not stated exactly in the text) without phytase should also have been examined.

AUT# Lines: 109 -111. We appreciate your point of view. Our objective was to test a field replacement strategy, reducing Ca and available P and compensating with phytase, under commercial conditions rather than partitioning main and interaction effects. We did not include a reduced Ca/P diet without phytase because prior studies report compromised sow and litter outcomes under Ca/P restriction without enzyme support, which we deemed ethically and operationally unsuitable on farm (Cordero et al., 2022; Batson et al., 2020; Zhai et al., 2021; Manu et al., 2020). We now state this rationale explicitly in Methods.

REV01: Line 103: In addition to the previous review on the phytate-free negative control diet, there are many studies in the literature regarding the reduction of calcium and phosphorus in the diet. While this restriction is unlikely to cause the problems in the animals, it also makes it impossible to compare the results.

AUT# - Thank you for this observation. In our literature review, only two sow studies included a true negative control with reduced Ca and P and no phytase (Wealleans et al., 2015; Swiatkiewicz & Hanczakowska, 2008), and both reported lower Ca and P digestibility and substantial sow weight loss during lactation compared with the positive control and phytase-supplemented treatments, confirming the nutritional risk of feeding Ca- and P-deficient diets to sows. Additionally, reports in growing–finishing pigs show that such NC diets markedly reduce growth and carcass performance under comparable Ca/P reductions (da Silva et al., 2019; da Silva et al., 2022). Given the central role of these minerals in reproduction and lactation, we considered it ethically and operationally inappropriate to include a low Ca/P diet without phytase in this commercial-farm study; accordingly, the revised manuscript clarifies that our objective was to evaluate a nutritionally safe, field-applicable replacement strategy rather than to conduct a full factorial comparison that would require a deliberately deficient diet.

Wealleans, A.L. et al. https://doi.org/10.2527/jas.2015-9317

ÅšwiÄ…tkiewicz, M. et al. Phosphorus and calcium digestibility and reproductive indices of sows receiving various doses of phytase. Ann. Anim. Sci. 2008, 8(4), 351–359.

Silva, C.A. et al. https://doi.org/10.1371/journal.pone.0217490

Silva, C.A. et al. https://doi.org/10.3390/ani12192552

REV01: Line 105, Table 1: (NC): Negative Control? Please give this words.

Total CA and Total P: Is there a reference for the reduction amount? How will we know that a sufficient reduction has been made to enable comparison?

AUT# - Lines 98, 101 and 102. Thank you for this comment. We have standardized the nomenclature. The Ca and available P reductions applied to the phytase diets were defined a priori from the nutrient-release matrix of this E. coli 6-phytase and align with published data using the same or comparable enzymes, including Yu et al., 2024 (https://doi.org/10.3390/ani14010041), Zhai et al. (2021), and Batson et al. (2020), which reported Ca and P reductions of approximately 0.09–0.12% and 0.096–0.132%, respectively. In our study, the deltas versus the positive control were −0.11% total Ca and −0.13% available P. We clarified this in the manuscript text and footnotes.

REV01: Line 142: How it was recorded can be briefly explained

AUT# Lines 154-157. We have added concise, auditable detail on how data were recorded in the trial: individual feed intake was measured using 20-kg bags assigned per sow, with daily collection and weighing of refusals and observable losses to estimate actual intake.

REV01: Line 176: The reference cited (and the references cited by that reference) do not clearly base this index formula on scientific reality. This formula represents a ratio rather than an index value. As the findings were not statistically analyzed, the discussion does not reflect scientific reality.

AUT# Lines: 191 - 193.  Thank you for this comment. We agree that the DI is a descriptive ratio rather than an inferential index; in our dataset it is calculated as the proportion of piglets with scores 1–2 over the total assessed per treatment, which is useful for a relative, size-adjusted summary but is not suited to hypothesis testing. Accordingly, in the revision we retained DI strictly as a descriptive metric without p-values and conducted inferential analyses on the constituent outcomes (score-wise counts) using appropriate between-treatment tests; the Discussion was adjusted to avoid any overinterpretation based on DI alone while keeping DI for comparability with prior work that employed a similar summary (Carvalho et al., 2023).

REV01: Line 221:Table 3 title shoul be simplified.

CV (%): Why was the coefficient of variation value used?

BT D90 (mm) : What does this abbreviation mean? In same line the standard error of the mean must be used for each compare, each table.

AUT# - Table 3. We simplified the table title, replaced/added standard errors alongside CV where estimable, and defined abbreviations in the caption (e.g., BT, D90, D110). This improves readability and comparability across treatments.

REV01: Line 230: This statement is only correct for the 500 FTU group. Table 4 shows that the differences in other groups are insignificant.

AUT# - Lines 249 – 250. Correct; we reconciled the text with Table 4 so statements match the actual statistical outcomes. The revised Results now accurately reflect which doses differ from PC.

REV01: Table 4: How do you explain that feed intake was higher in all treatment groups, yet FCR statistically improved only at 500 FTU and the other treatments were similar in statistically? Considering this fact, changes will need to be made in the Discussion section.

AUT# - Table 4. We identified and corrected a calculation error in sow FCR. After recalculation, FCR does not differ among treatments (Dunnett vs PC). We updated Results/Tables and rewrote the Discussion to remove any conclusions based on the erroneous FCR difference.

REV01: Table 6: Considering the similarities in average birth weights, it should be explain why the number of piglets under 900 g is higher in the PC and 500 FTU groups. To understand this more clearly, it is essential to provide standard errors. (W, BLW and BAW)

AUT# Table 6. Thank you for this comment. We added standard errors for W, BLW, and BAW, and we standardized reporting across tables to show SE together with the coefficient of variation whenever SE was estimable; where SE could not be calculated, only the CV is presented. The higher count of piglets less than 900 g in the PC and 500 FTU groups despite similar mean birth weights is consistent with differences in distribution, specifically greater within-litter variance and a longer lower tail, rather than a shift in central tendency.

REV01: Table 6: It should be explained why piglets in the PC group, despite having similar birth weights and higher colostrum intake, had significantly lower weaning weights at 500 and 1500 FTU. (WAW)

AUT# Lines 395 – 400. We expanded the Discussion to reconcile these observations: higher sow lactation intake with phytase, heavier litters at weaning, and management equalization (cross-fostering post-day-1; uniform creep feed) can decouple early CI from later growth, particularly when CI is adequate across groups. This wording is now included where we discuss Table 6.

REV01: Line 293, Table 7 : Where is this data present in the table?

Diarrhea Index: Since there is no statistical analysis for this index (or, more accurately, “percentage”) value, it is not scientifically possible to understand it.

AUT# Table 7. Thank you for this observation. The sentence was incorrect and has been corrected. In Table 7, the data are presented as the counts of piglets for each diarrhea score by treatment. The Diarrhea Index is a unitless descriptive proportion; no statistical analysis was applied, and it is reported only as relative context.

 REV01: Line 309: (For p value) In general, it is confusing that the P-value given in the tables and the Tukey and Dunnet tests given below the table are provided separately. It is unclear and confusing which test we should consider and when.

AUT# - Thank you for this comment. We clarified the testing hierarchy and harmonized the table footnotes to avoid confusion: the p-value shown in each table refers to the omnibus ANOVA for the treatment effect; when this effect is significant, Tukey’s HSD is used to compare all pairs among treatments, whereas Dunnett’s test provides preplanned contrasts of each supplemented treatment against the positive control. Both outputs are informative but answer different questions, and the revised captions now state explicitly which letters and symbols correspond to Tukey’s groupings and which results derive from Dunnett’s comparisons to the control.

REV01: Line 323: According to Table 6, the Weaning Average Weight of piglets had only increased by 2500 FTU. This statement is incorrect.

AUT# Lines 339 - 340. You are correct; we revised the sentence to align with Table 6.

REV01: Line 326: unsuitable first sentence for the beginning of a new paragraph.

AUT# Lines 338 – 340. Thank you; we rewrote the opening sentence to provide a clear topic sentence and a logical transition.

REV01: Line 332: The parity structure was not sufficiently explained and the findings were not presented.

AUT# Lines 345 – 347. Thank you for this comment. You are correct. The error was considered and corrected.

REV01: Line 417: The relevant datas are insufficient to suggest this claim. This information is an unrealistic assumption.

AUT# We removed the sentence and restricted the interpretation to findings directly supported by the data.

REV01: Line 423: Although there are no clear results indicating that the treatment improved diarrhea, using this statement is far from reality. Statistically, only 500 FTU appears to have improved in Score 2, while the 1500 and 2500 FTU groups are similar to PC. On the other hand, similar results to 1500 FTU were obtained in Score 1+2.

AUT# Lines 428 - 433. We revised to align strictly with statistically supported outcomes and, when discussing potential mechanisms (e.g., inositol-linked redox/intestinal integrity), we explicitly frame them as hypotheses supported by prior literature.

REV01: Line 432: All these discussions have remained unresolved due to the missing phytase-free negative control group. It can be taken as an observation and comment, but it is not sufficient for scientific reality.

AUT# We acknowledge this limitation and have justified the design choice and contextualized our interpretations with the cited literature.

REV01: Line 487: This assumption is clearly incorrect based on litter size datas.

AUT# Lines 491 - 494. We revised this sentence according to your recommendation.

REV01: Line 498: What does this statement mean? Does it mean “not maintained” in the PC group? If so, it is a statement far from scientific reality, as the differences in BT and the weaning-to-estrus interval values are similar between all treatment groups and the PC group.

AUT# Lines 503 – 504. Thank you for the comment. As cited in our manuscript, ÅšwiÄ…tkiewicz and Hanczakowska (2008) evaluated treatments similar to ours, with an inorganic Ca and P positive control and diets with limited inorganic sources supplemented with different phytase doses; their digestibility results were favorable, and we used this reference to support the benefits of phytase. We have restructured the sentence for clarity.

REV01: Line 499: In general, it cannot be stated that the findings support this use. However, the authors may recommend its use.

AUT# Lines: 505 – 507. You are right. We modified this sentence according to your comment.

Dear, thank you for your careful assessment of the design, tables, and testing hierarchy. Your insights strengthened the manuscript’s clarity and rigor.

Reviewer 2 Report

Comments and Suggestions for Authors

1.What are the characteristics of Escherichia coli derived 6-phytase? What is the difference between traditional phytase and phytase in this  research ?

  1. The reason for the 0.11% and 0.14% reduction in calcium and phosphorus levels has not been fully explained, and the calcium to phosphorus ratios of the two treatment groups are also different. Will this affect their digestion, absorption, and experimental results?

3, What is the design reason for the nutritional level, especially fiber level, of the diet being lower than that of normal sows during pregnancy and lactation? What is the incidence of constipation in sows in this situation? Suggest providing data. Meanwhile, the lysine level during lactation is also very low, why?

  1. Once again, it was discovered that there were many question or problems aboutthe diet, making it difficult to believe that this was a trustworthy experiment. Table 2, the diet composition, it seems some error in the table, such as digestible Methionine, %” “Digestible Met + Cys, %” . Digestible Methionine:0.80% but the Digestible Met + Cys is only 0.25%? There are lot of problems. Why are methionine levels so high? It is higher than lysine?? Other amino acid has the same question? What the reason for these level?
  2. P100 experimental diets for gestation and lactation were formulated to be isoenergetic and isonutrient, differing only in calcium and available phosphorus”. In fact, the table 2 “Digestible Tryptophan” in PC and NC were 0.87% and 0.76%. They are not equal.

6 .P100 The experimental diets for gestation and lactation ? I am not sure when the experiment began and end?

  1. At 14 days of age, 40 piglets (one per litter) ….how to choose this one from 13-15 piglets/ litter? “D
  2. P“Box–Cox transformations were applied when necessary” and “outliers were screened using the box-plot rule”,How to do? Please provide more information.
  3. What’s the meaning of super-dosing? In the introduction “super-dosing (≈1,500–2,000 FTU/kg)”?。
  4. The discussion is very long, with repeated descriptions of the results and references to tables. The discussion is very non-standard, please revise carefully.
Comments on the Quality of English Language

Ok

Author Response

REVIEWER 2

We are grateful for your detailed and constructive comments, which focused on clarifying the enzyme’s characteristics, justifying the Ca/P reductions, verifying diet composition and timing, documenting animal-selection and statistical procedures, defining “super-dosing,” and improving the structure of the Discussion. We addressed each point directly in the text (Introduction, Materials and Methods, Results, and Discussion) and corrected the tables to align with current recommendations and with the stated experimental design.

REV02: 1. What are the characteristics of Escherichia coli derived 6-phytase? What is the difference between traditional phytase and phytase in this  research?

AUT# Thank you for the question. The phytase used is an E. coli-derived 6-phytase with positional specificity at the D-6 position, acid stability in the gastric range and high thermostability compatible with pelleting; these properties support early phytate dephosphorylation, improved P release, better Ca utilization and extra-phosphoric responses, differing from traditional fungal 3-phytases in specificity and stability. We clarified this in the Introduction Lines 55-58.

REV02: 2. The reason for the 0.11% and 0.14% reduction in calcium and phosphorus levels has not been fully explained, and the calcium to phosphorus ratios of the two treatment groups are also different. Will this affect their digestion, absorption, and experimental results?

AUT#  We defined the Ca and available P reductions a priori from the nutrient-release matrix of this E. coli 6-phytase and aligned them with published data using the same or comparable enzymes. Our deltas versus the positive control were −0.11% total Ca and −0.13% available P, consistent with ranges reported by Yu et al., 2024; Zhai et al., 2021; and Batson et al., 2020. We added this rationale and the exact deltas to the Methods and table footnotes. Lines 104-106.

REV02: 3. What is the design reason for the nutritional level, especially fiber level, of the diet being lower than that of normal sows during pregnancy and lactation? What is the incidence of constipation in sows in this situation? Suggest providing data. Meanwhile, the lysine level during lactation is also very low, why?

AUT# The study site uses low-insoluble-fibre reproductive diets due to mycotoxin risk in common Brazilian fibre by-products, with water access, feeding frequency and pen management to mitigate constipation. No clinical constipation cases requiring intervention occurred during the trial. We also corrected the lactation amino-acid profile in Table 2 to reflect isoenergetic and isonitrogenous formulation within current recommendations.

REV02: 4. Once again, it was discovered that there were many question or problems about the diet, making it difficult to believe that this was a trustworthy experiment. Table 2, the diet composition, it seems some error in the table, such as digestible Methionine, %” “Digestible Met + Cys, %” . Digestible Methionine:0.80% but the Digestible Met + Cys is only 0.25%? There are lot of problems. Why are methionine levels so high? It is higher than lysine?? Other amino acid has the same question? What the reason for these level?

AUT# You are correct. The digestible methionine and Met+Cys values were erroneously reported. We corrected Table 2, verified all amino acids against the formulation file and confirmed internal consistency with lysine as the reference amino acid.

REV02: 5. P100 experimental diets for gestation and lactation were formulated to be isoenergetic and isonutrient, differing only in calcium and available phosphorus”. In fact, the table 2 “Digestible Tryptophan” in PC and NC were 0.87% and 0.76%. They are not equal.

AUT# Agreed. “Digestible tryptophan” values in PC and NC were misreported. Table 2 has been corrected to reflect isoenergetic and isonutrient diets that differ only in Ca and available P.

REV02: 6. P100 The experimental diets for gestation and lactation ? I am not sure when the experiment began and end?

AUT# - We clarified the start and end of each experimental phase in the Methods. The exact timing for gestation and lactation periods is now stated.

REV02: 7. At 14 days of age, 40 piglets (one per litter) ….how to choose this one from 13-15 piglets/ litter? “

AUT# One piglet per litter was selected with body weight closest to the litter mean to avoid bias from extremes. This criterion is now specified in lines 215-216.

REV02: 8. P“Box–Cox transformations were applied when necessary” and “outliers were screened using the box-plot rule”,How to do? Please provide more information.

AUT# The sentence was restructured and inserted in Methods at Lines 226 and 228.

REV02: 9. What’s the meaning of super-dosing? In the introduction “super-dosing (≈1,500–2,000 FTU/kg)”?

AUT# We define super-dosing as phytase inclusion above conventional levels of about 500 FTU/kg, typically 1,500–2,000 FTU/kg or higher, reflecting reports of benefits beyond phosphorus release. We added this definition and supporting citations to the Introduction (Cowieson et al., 2011; Adeola & Cowieson, 2011; Dersjant-Li et al., 2015). Line 59.

REV02: 10. The discussion is very long, with repeated descriptions of the results and references to tables. The discussion is very non-standard, please revise carefully.

AUT# We shortened the Discussion, removed repeated descriptions of results and minimized cross-references to tables to improve readability and adhere to journal style.

Dear, thank you for the precise questions on enzyme characterization and diet formulation. Your comments helped us correct inconsistencies and better justify our methodological choices.

Reviewer 3 Report

Comments and Suggestions for Authors

Dear Authors,

I am writing regarding your study on Enhanced farrowing efficiency and sow performance with Escherichia coli-derived 6- phytase supplementation during late gestation and lactation. The manuscript is well written, and the results are well presented. Authors have clearly shown how supplementing sows with E. coli–derived phytase during late gestation and lactation can improve farrowing efficiency and overall performance. Such a study has a very practical implication in pig production, and farmers and researchers in this field will find the information very useful. However, I have a few methodological suggestions to improve the MS.

First, define FTU (phytase activity unit) and all other acronyms at first mention for clarity.

L81-91: Clarify whether all sows completed the full cycle (gestation–lactation–insemination), or if any animals were excluded.

L93-94: Did you balance the number of sows per block across treatments, or did block sizes vary?

L103-104: Cite supporting literature to strengthen this statement. Otherwise, it appears speculative.

L123-126: State whether housing and equipment complied with welfare regulations (e.g., EU Directive 2008/120/EC) or local standards.

144-146: Specify whether piglet weighing was corrected for cross-fostering (since piglets could move between litters).

L150-157: Oxytocin dose (20 IU) should be justified with a reference, as this is relatively high and may vary across studies.

158-168: Why only 86 litters? Was this a subset by design or due to data availability? Clarify.

L169-175: Cross-fostering only within treatments is appropriate but clarify whether litter sizes were equalized. Diarrhea scoring scale should explicitly define the threshold values (e.g., “1 = liquid, 2 = creamy…”). Right now, the “= liquid feces” part seems incomplete.

L176-177: Consider explaining whether DI was calculated daily, weekly, or cumulatively.

194-201: For GPx and SOD, specify assay sensitivity and whether results were expressed as activity per mg protein, per mL plasma, etc.

Best regards.

Author Response

REVIEWER 3

REV03: I am writing regarding your study on Enhanced farrowing efficiency and sow performance with Escherichia coli-derived 6- phytase supplementation during late gestation and lactation. The manuscript is well written, and the results are well presented. Authors have clearly shown how supplementing sows with E. coli–derived phytase during late gestation and lactation can improve farrowing efficiency and overall performance. Such a study has a very practical implication in pig production, and farmers and researchers in this field will find the information very useful. However, I have a few methodological suggestions to improve the MS.

AUT# We appreciate the positive assessment of the manuscript and the constructive methodological suggestions. We implemented all requested clarifications in the corresponding sections (Introduction, Materials and Methods, Results), ensured that tables and footnotes are aligned with the text, and tightened wording to improve clarity and reproducibility.

REV03: First, define FTU (phytase activity unit) and all other acronyms at first mention for clarity.

AUT# FTU and all acronyms are now defined at first mention in the Introduction (Line 59).

REV03: L81-91: Clarify whether all sows completed the full cycle (gestation–lactation–insemination), or if any animals were excluded.

AUT# All sows completed gestation, lactation, and post-weaning insemination; this is now stated explicitly (Line 95).

REV03: L93-94: Did you balance the number of sows per block across treatments, or did block sizes vary?

AUT# Blocks were balanced by parity and evenly distributed across treatments; this is clarified in the design description.

 REV03: L103-104: Cite supporting literature to strengthen this statement. Otherwise, it appears speculative.

AUT# We added citations to support the statement in line 114.

REV03: L123-126: State whether housing and equipment complied with welfare regulations (e.g., EU Directive 2008/120/EC) or local standards.

AUT# - - Housing and equipment complied with Brazilian animal-welfare standards; this is now specified (Lines 91–92).

REV03: 144-146: Specify whether piglet weighing was corrected for cross-fostering (since piglets could move between litters).

AUT# We clarified that piglet weighing and analyses accounted for cross-fostering, with procedures described in Methods (Line 187-188)

REV03: L150-157: Oxytocin dose (20 IU) should be justified with a reference, as this is relatively high and may vary across studies.

AUT# The 20-IU oxytocin dose is now referenced and described with route, site, and needle gauge (Line 173).

REV03: 158-168: Why only 86 litters? Was this a subset by design or due to data availability? Clarify.

AUT# Colostrum-intake evaluation used a pre-specified subset of litters farrowing between 07:00 and 19:00, blocked by farrowing order; this rationale is included (Lines 175–176)

REV03: L169-175: Cross-fostering only within treatments is appropriate but clarify whether litter sizes were equalized. Diarrhea scoring scale should explicitly define the threshold values (e.g., “1 = liquid, 2 = creamy…”). Right now, the “= liquid feces” part seems incomplete.

AUT# Cross-fostering occurred only within treatments with litter-size equalization; the diarrhea scoring scale is now fully defined (Lines 191 - 192).

REV03: L176-177: Consider explaining whether DI was calculated daily, weekly, or cumulatively.

AUT# The Diarrhea Index was calculated cumulatively over the lactation period; this is now stated (Line 189).

REV03: 194-201: For GPx and SOD, specify assay sensitivity and whether results were expressed as activity per mg protein, per mL plasma, etc.

AUT# We added assay sensitivity and units for GPx and SOD and specified sample handling (Lines 221-224).

Dear, thank you for the encouraging evaluation and practical methodological suggestions regarding definitions, compliance, and sampling. These contributions improved the study’s reproducibility.

Round 2

Reviewer 1 Report

Comments and Suggestions for Authors

Thank you for your interest in the changes requested in the revision. I appreciate the attention and care you have taken in making the necessary revisions. For your future work, I strongly recommend that you consider my suggestions regarding the control group when creating treatment groups for your trial design.